# Manchester Intermittent Diet in Gestational Diabetes Acceptability Study (MIDDAS-GDM): a two-arm randomised feasibility protocol trial of an intermittent low-energy diet (ILED) in women with gestational diabetes and obesity in Greater Manchester

Elizabeth Dapre [1,2] Basil G Issa,[2,3] Michelle Harvie [2,4,5] Ting-Li Su,[6]
Brian McMillan [7] Andrea Pilkington,[2] Fahmy Hanna,[8] Avni Vyas,[2,9]
Sarah Mackie,[2] James Yates,[2] Benjamin Evans,[2] Womba Mubita,[3]
Cheryl Lombardelli[2,5]

For numbered affiliations see end of article.

**Correspondence to**
Dr Elizabeth Dapre;
elizabeth.dapre@manchester.ac.uk

## ABSTRACT

**Introduction** The prevalence of gestational diabetes mellitus (GDM) is rising in the UK and is associated with maternal and neonatal complications. National Institute for Health and Care Excellence guidance advises first-line management with healthy eating and physical activity which is only moderately effective for achieving glycaemic targets. Approximately 30% of women require medication with metformin and/or insulin. There is currently no strong evidence base for any particular dietary regimen to improve outcomes in GDM. Intermittent low-energy diets (ILEDs) are associated with improved glycaemic control and reduced insulin resistance in type 2 diabetes and could be a viable option in the management of GDM. This study aims to test the safety, feasibility and acceptability of an ILED intervention among women with GDM compared with best National Health Service (NHS) care.

**Method and analysis** We aim to recruit 48 women with GDM diagnosed between 24 and 30 weeks gestation from antenatal clinics at Wythenshawe and St Mary's hospitals, Manchester Foundation Trust, over 13 months starting in November 2022. Participants will be randomised (1:1) to ILED (2 low-energy diet days/week of 1000 kcal and 5 days/week of the best NHS care healthy diet and physical activity advice) or best NHS care 7 days/week until delivery of their baby. Primary outcomes include uptake and retention of participants to the trial and adherence to both dietary interventions. Safety outcomes will include birth weight, gestational age at delivery, neonatal hypoglycaemic episodes requiring intervention, neonatal hyperbilirubinaemia, admission to special care baby unit or neonatal intensive care unit, stillbirths, the percentage of women with hypoglycaemic episodes requiring third-party assistance, and significant maternal ketonaemia (defined as ≥1.0 mmol/L). Secondary outcomes will assess the fidelity of delivery of the interventions, and qualitative analysis of participant and healthcare professionals' experiences of the diet. Exploratory outcomes include the number of women requiring metformin and/or insulin.

**Ethics and dissemination** Ethical approval has been granted by the Cambridge East Research Ethics Committee (22/EE/0119). Findings will be disseminated via publication in peer-reviewed journals, conference presentations and shared with diabetes charitable bodies and organisations in the UK, such as Diabetes UK and the Association of British Clinical Diabetologists.

**Trial registration number** NCT05344066.

### STRENGTHS AND LIMITATIONS OF THIS STUDY

⇒ This mixed-method feasibility study includes both quantitative and qualitative evaluation of the acceptability of the dietary intervention.
⇒ This study has been informed by an experienced patient and public involvement and engagement group.
⇒ This study involves a small sample size and is not powered to show efficacy of the intervention.
⇒ Women joining this study are likely to be highly motivated and adherence may not reflect that seen in the wider general population.

## INTRODUCTION
### Background

In the UK up to 16% of pregnant women develop gestational diabetes mellitus (GDM) and the incidence is rising, in part due to increasing rates of obesity and maternal age.[1 2] GDM is associated with maternal and neonatal complications (the risk increases

with poor glycaemic control), including macrosomia, shoulder dystocia, caesarean sections, neonatal hypoglycaemia and/or hyperbilirubinaemia, preterm delivery, pre-eclampsia and stillbirth.[2] Women who have had GDM have an estimated 7-fold to 10-fold risk of developing type 2 diabetes (T2DM) later in life, and their children have a higher risk of developing adult obesity and T2DM.[2–4]

Excessive weight gain in pregnancy is a particular problem for women with GDM.[5] Harper et al demonstrated that, in women with GDM, every additional 1 lb/week gained following diagnosis of GDM resulted in a 36%–83% increased risk of pre-eclampsia, caesarean section, macrosomia and large for gestational age babies.[5] Such studies highlight the importance of adequate weight control throughout pregnancy in women with GDM in order to reduce both maternal and neonatal complications.

First-line therapy for GDM is diet and physical activity. National Institute for Health and Care Excellence (NICE) guidance encourages a healthy diet with increased fruit and vegetables, low-glycaemic index (GI) foods, reduced refined sugars, regular mealtimes and regular physical activity.[6 7] These dietary measures fail to achieve glycaemic targets in ~30% of women who require medication with metformin and/or insulin.[8] A range of dietary approaches have been studied including daily diets promoting low-GI diets (limiting refined and promoting complex carbohydrates), continuous modest energy restriction (1800 kcal/day) and low carbohydrate diets.[9] There is currently no strong evidence base for any particular dietary regimen to improve outcomes in GDM.

### Intermittent low-energy diets (ILED)
The pathogenesis of GDM is strongly linked to obesity and chronic insulin resistance with many clinicians viewing GDM as a form of evolving T2DM. ILEDs typically include several days of a food based or meal replacement (eg, drinks/bars) low-energy diet (650–1000 kcal), with a standard healthy (non-restrictive) diet recommended on the remaining days of the week. These diets are associated with significant reductions in weight, insulin resistance and hyperglycaemia in patients with pre-diabetes (Haemoglobin A1C (HbA1c) between 42 and 47 mmol/mol, impaired glucose tolerance, or impaired fasting glycaemia), those with T2DM, and otherwise healthy subjects with overweight/obesity.[10–17] These changes are equivalent to, or greater than, those achieved with standard daily energy restriction. A popular intermittent diet involves 2 consecutive or non-consecutive days/week of a low-energy diet (650–1000 kcal) and 5 days of normal eating/week, known as the 5:2 diet. The Manchester Intermittent versus Daily Diabetes App Study (MIDDAS), a study comparing an ILED and a continuous low-energy diet in T2D conducted in our unit, has shown the feasibility and safety of an ILED (800 kcal for 2 days/week) in patients with T2DM and obesity, including those using insulin.[18] At the end of the study, approximately 70% of participants in the ILED group completed the study and achieved a 6% reduction in their baseline body weight. 42% achieved an HbA1c of <48 mmol/mol.[18] Given the strong overlap between GDM and T2DM, an ILED may be a promising dietary intervention for those with GDM.

A successful dietary approach to glycaemic control could empower women to take charge of the management of their GDM. Women with GDM are motivated to modify their diet driven by a desire to improve foetal outcomes.[19–21]

Our patient and public involvement and engagement (PPIE) work indicates that women find the current NICE healthy eating guidance[6 7] confusing and vague. Our PPIE work has indicated that women are keen to try alternative dietary approaches, particularly if alternative diets are more effective in preventing the need to progress to medications such as metformin and insulin.

### Aim
The aim of this trial is to test the safety, feasibility and acceptability of an ILED in GDM to inform a future large-scale randomised controlled trial (RCT).

## METHODS
### Trial design
The study is a 28-week feasibility two-arm RCT in one National Health Service (NHS) trust performed in patients with GDM and body mass index (BMI) ≥27.5 kg/m$^2$, or ≥25 kg/m$^2$ in high-risk minority ethnic groups (ie, South Asian, Black African and African Caribbean) in Greater Manchester between December 2022 and July 2024.[22 23] There will be an embedded qualitative substudy for participants and healthcare professionals (HCPs). Due to the nature of the intervention, it will not be possible to blind the participants, clinicians or study team to the treatment allocation after randomisation (the statistician and laboratory technicians will remain blinded).

### Trial setting and recruitment
Participants will be recruited from antenatal clinics at Wythenshawe and St Mary's Hospitals, Manchester Foundation Trust (MFT) between November 2022 and December 2023. This is an urban area within Greater Manchester, and MFT serves patients from a wide range of minority ethnic and sociodemographic backgrounds. Women may self-refer to the antenatal clinic or be referred by their primary care team. Assessments will be carried out at MFT, or remotely if required by COVID-19 restrictions. The qualitative substudy will be carried out at MFT, remotely, or at a location of the participant's choosing. We aim to recruit eligible participants over a period of 13 months. Potential participants will be given written information about the study and the opportunity to ask questions about the study prior to providing written consent (online supplemental files 1 and 2).

## Eligibility criteria

### Participant flow

Participants who fulfil the broad eligibility criteria will be notified about the trial by the GDM nurse/midwife at the time of their diagnosis. Those who are interested will be provided with a comprehensive patient information sheet (online supplemental file 1) and more detailed eligibility screening questions (figure 1). They will be asked to attend their first appointment having fasted for at least 6 hours and complete a 4-day food diary (in line with our department's usual care). On attending their first routine clinic appointment, interested participants will receive further information from the research team. They will have the opportunity to ask questions, have their eligibility confirmed and will be asked for their written consent to take part. Baseline assessments will be taken and participants will be randomised to their allocated treatment group using an online randomisation platform. Participant flow through the study is demonstrated in figure 2.

### Sample size

We plan to recruit 24 participants per study arm (n=48) which, when considering an estimated attrition rate of 15%, will provide complete outcome data on 40 participants.[24–26] It has been estimated that 24 participants per group will be sufficient to determine study outcomes, in line with sample size recommendations for feasibility studies.[27–29]

This number will allow us to enable estimation of recruitment/retention parameters with sufficient precision. For example, based on 40 completed participants, it will enable recruitment rates in the region of 25% to be estimated with an error of ±13.42% at most; retention of 85% will be estimated with error of ±11.07% at most. It is also sufficient for estimation of variability (eg, SD) in gestational weight gain and capillary glucose concentrations (proposed outcomes for the future definitive trial) with negligible bias.[30]

### Randomisation

The randomisation schedule will be independently set up and known only by the trial statistician. The trial statistician will be blinded to the participant's identity using 'sealed envelope' software (https://www.sealedenvelope.com/). Randomisation will be carried out by generating an online pseudorandom list with random permuted blocks of varying size, known only to the statistician, and will be stratified for two variables:

► Age (18–35, >35 years).
► BMI (27.5–34.99 kg/m² and>35 kg/m²; >25–32.49 kg/m² and >32.5 kg/m² for high-risk minority ethnic groups (ie, South Asian, Black African and African Caribbean).

**Inclusion Criteria**

➢ Pregnant women ≥18 years
➢ BMI of ≥27.5kg/m2 or a BMI ≥25 kg/m² in high risk minority ethnic group (i.e. South Asian, Black African, African Caribbean) and <50 kg/m2 at booking appointment (8-12 weeks' gestation)
➢ Newly diagnosed GDM according to local diagnostic criteria (fasting glucose ≥5.3mmol/l and/or 2-hour postprandial glucose ≥8.5mmol/l in a 75g OGTT) scheduled to receive first line diet and physical activity (best NHS care)
➢ 24-30 weeks' pregnant at screening appointment

**Exclusion Criteria**

➢ Pregestational type 1 or type 2 diabetes.
➢ Fasting glucose of ≥7 or 2-hour postprandial of ≥11 on OGTT (immediate intervention with medication would be required in this group of women)
➢ Current multiple pregnancy
➢ Maturity Onset Diabetes of the Young (MODY)
➢ Significant comorbid disease that in PI's opinion would preclude participation in the study e.g. chronic kidney disease, significant cardiac disease, significant history of disordered eating or severe psychological problems.
➢ Current participation in a GDM medication treatment trial
➢ People who are not capable of providing informed consent or adhering to the monitoring and safety protocols
➢ People who have previously had bariatric surgery for weight loss including gastric bypass and sleeve gastrectomy, and/or those prescribed weight loss medications (e.g. orlistat).
➢ Medications at the time of the OGTT that may interfere with results (e.g. high dose oral steroids, immunosuppressants)
➢ Previous history of intrauterine growth restriction
➢ Women who have lost more than 5% of their weight from booking appointment to screening appointment.

**Figure 1** Inclusion and exclusion criteria. BMI, body mass index; GDM, gestational diabetes; NHS, National Health Service; OGTT, oral glucose tolerance tests.

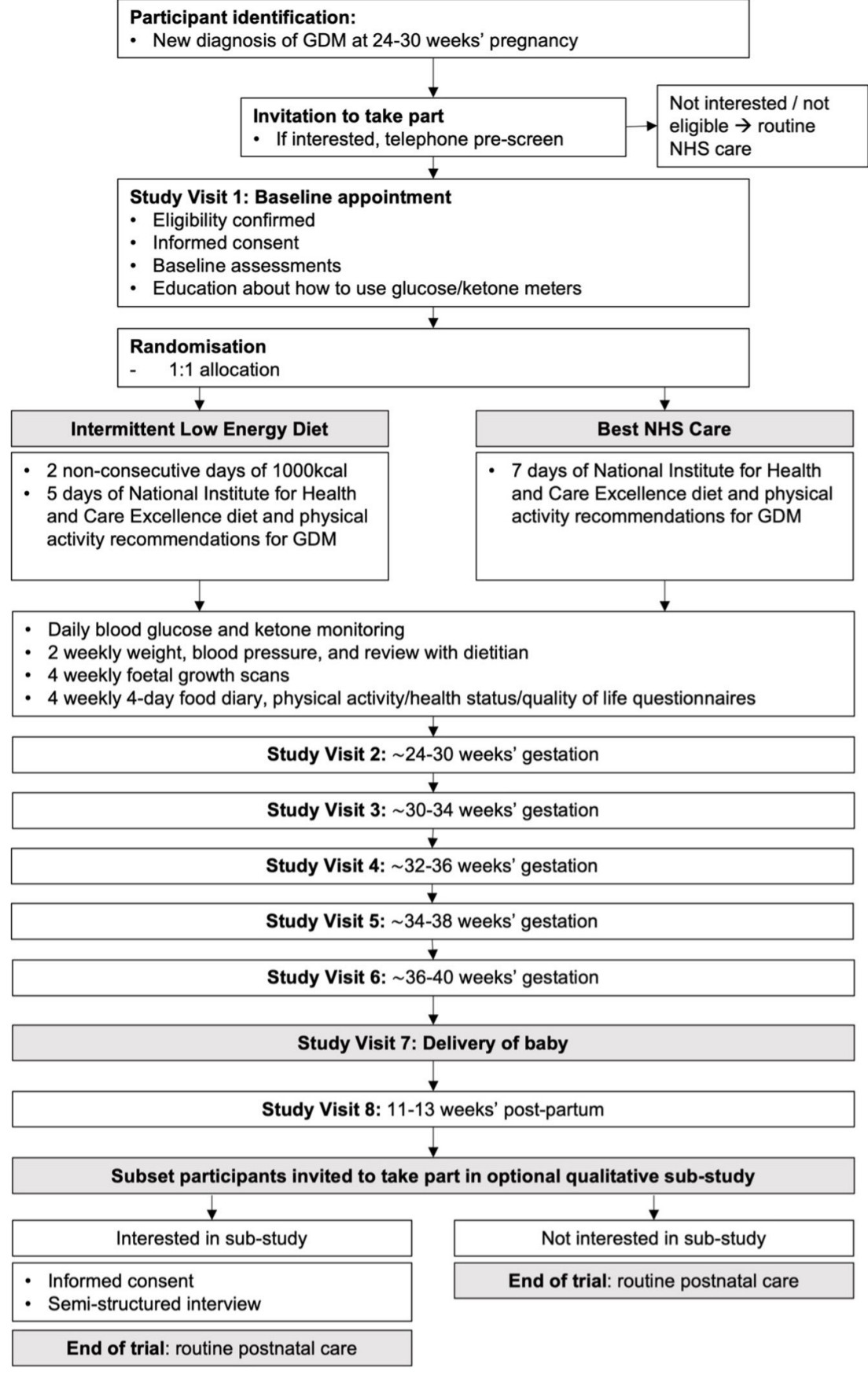

**Participant identification:**
• New diagnosis of GDM at 24-30 weeks' pregnancy

**Invitation to take part**
• If interested, telephone pre-screen

Not interested / not eligible → routine NHS care

**Study Visit 1: Baseline appointment**
• Eligibility confirmed
• Informed consent
• Baseline assessments
• Education about how to use glucose/ketone meters

**Randomisation**
- 1:1 allocation

**Intermittent Low Energy Diet**
• 2 non-consecutive days of 1000kcal
• 5 days of National Institute for Health and Care Excellence diet and physical activity recommendations for GDM

**Best NHS Care**
• 7 days of National Institute for Health and Care Excellence diet and physical activity recommendations for GDM

• Daily blood glucose and ketone monitoring
• 2 weekly weight, blood pressure, and review with dietitian
• 4 weekly foetal growth scans
• 4 weekly 4-day food diary, physical activity/health status/quality of life questionnaires

**Study Visit 2:** ~24-30 weeks' gestation

**Study Visit 3:** ~30-34 weeks' gestation

**Study Visit 4:** ~32-36 weeks' gestation

**Study Visit 5:** ~34-38 weeks' gestation

**Study Visit 6:** ~36-40 weeks' gestation

**Study Visit 7: Delivery of baby**

**Study Visit 8:** 11-13 weeks' post-partum

**Subset participants invited to take part in optional qualitative sub-study**

Interested in sub-study

Not interested in sub-study

• Informed consent
• Semi-structured interview

**End of trial**: routine postnatal care

**End of trial**: routine postnatal care

**Figure 2** Participant flow through trial. GDM, gestational diabetes; NHS, National Health Service

These stratification variables have been chosen to reduce potential bias as we expect varying severity of GDM with increasing age and BMI and possible differences in diet adherence.[31]

Treatment to intervention and control groups will be allocated in a 1:1 ratio. A member of the research team who will be unaware of the randomisation algorithm (principal investigator, clinical research nurse, clinical research fellow or project manager) will trigger the randomisation procedure onsite; participants and clinicians will then be informed of the allocated treatment group. Clinicians will not be blinded due to the need to remain astute to safety, adherence and side effects, requiring open and honest discussions with patients at each appointment. The statistician will remain blinded to treatment allocation until all outcome measures for all subjects have been collected.

## Interventions

### Study arm 1: best NHS care diet

All dietetic advice will be face-to-face or via video calls or the telephone. Participants will receive one-to-one personalised written and verbal advice from a dietitian to follow NICE diet and physical activity recommendations.[6 7] Dietitians and midwives will receive training to ensure standardised delivery of information in clinic, and standardised patient information leaflets will be supplied to include information about increased fruit/vegetable intake, low-GI foods and a reduction in free sugars. Information will include advice about the importance of regular meals; dietary advice aims to ensure that participants include at least 70 g protein/28 g fibre, and predominantly monounsaturated and polyunsaturated fats as per American Diabetes Association recommendations.[32] Participants will be advised to be physically active, for example, walking for 30 min after a meal. Participants will receive ongoing dietetic education and support every 2 weeks until delivery. This level of support is higher than typically provided in NHS GDM antenatal clinics due to limited resources but has been used to reflect best NHS care. They will receive suggested menus and recipes to follow the NICE recommended healthy diet for GDM. Participants will be asked to measure their capillary glucose four times each day and their ketones on two random (recorded) days of the week of their choosing (online supplemental file 3).

### Study arm 2: ILED

Participants will receive the same level of dietetic support as the best NHS care group. They will be given advice on adopting an ILED which involves 2 non-consecutive low-energy diet days/week (1000 kcal to include 100 g low-GI carbohydrate and 70 g of protein) and 5 days/week of the NICE healthy eating low-GI diet and physical activity recommended for the best NHS care group. The low-energy days involve women selecting a set number of portions of protein, carbohydrate, fat, fruit, vegetables and dairy/dairy alternatives as described in previous studies.[33] Each low-energy day includes ~210 g of lean protein foods, 3–4 portions of wholegrain carbohydrates, 1×7 g portion of fat, 5 portions of vegetables, 2 of fruit and 3 of dairy/dairy alternatives. Food and drink will be self-selected and not provided by the study team. Participants will be provided with comprehensive food lists, advice on portion sizes for the low-energy days and suggested menus and recipes to follow for both the low-energy and NICE recommended healthy diet days (online supplemental file 4). Both diets can be successfully adapted for people of different ethnicities and those following omnivorous, vegetarian and vegan diets. Participants will be asked to measure their capillary glucose four times each day and their ketones on (and the morning after) the two low-energy days (online supplemental file 3).

## Outcomes

### Primary outcomes

► Uptake rate measured as a percentage of eligible participants who consent to take part, including the proportion of women who were screened who did not meet the eligibility criteria, and the number of women who did not give consent to take part.

► Recruitment rate measured as the number of eligible participants who consent to take part per month.

► Retention rate measured as the number of randomised participants who complete the trial (those who attend the final visit) and the percentage of participants who attend all eight visits.

► Adherence to the dietary interventions assessed from self-reported adherence to the potential low-calorie days between randomisation and delivery.

► Completion of self-assessed glucose and ketone readings assessed as a percentage of the required readings.

► Safety outcomes:
  – Percentage of women following ILED/best NHS care with hypoglycaemia (episodes of blood glucose of <3.0 mmol/mol) and hypoglycaemia requiring third-party assistance as measured by participants.
  – Percentage of women who develop significant ketonaemia in both groups (defined as ≥1.0 mmol/L) as measured by participants.
  – Percentage of neonatal hypoglycaemic episodes requiring intervention (blood glucose checked 2 hours post delivery and 2 hours thereafter for 12 hours according to local protocol), neonatal birth weight, gestational age at delivery, hyperbilirubinaemia/jaundice and/or admission to Special Care Baby Unit or neonatal intensive care, and stillbirths.
  – The incidence and rate of other adverse events (eg, headaches, lethargy, constipation or complications requiring hospital admission) between the start of the trial intervention and delivery recorded as mild, moderate and severe, as defined by Common Terminology Criteria for Adverse Events (CTCAE V.5).[34] Hospital admission for routine labour and delivery will not be classified as an adverse event.

## Secondary outcomes

- ► Completeness of collection of trial endpoints including the percentage of completed weight measurements, 4-day food diaries and International Physical Activity Questionnaire (IPAQ) scores.
- ► Fidelity of delivery of the interventions will be measured through the number and modality of completed planned patient contacts, electronic and paper food diaries, and self-reported capillary glucose and ketone measurements.
- ► Qualitative analysis of the acceptability and implementation of the interventions will be explored among a subset of participants (~10 in each group) and HCPs through in-depth interviews.

## Exploratory outcomes

The following outcomes will be explored without statistical inference.

- ► Maternal outcomes:
  - – The percentage of women requiring metformin and/or insulin.
  - – Four-point capillary glucose profiles during the third trimester (four times daily until delivery).
  - – Change in fasting blood test results between baseline measurements, 36–37 weeks gestation, and 12 weeks post delivery (including oral glucose tolerance tests).
  - – Mode of delivery, development of preeclampsia, polyhydramnios (maximum liquor volume pool depth ≥8 cm).
  - – Quality of life and health status questionnaires (World Health Organisation Quality of Life (Brief Version) (WHOQoL-BREF) and 36-Item Short Form Survey (SF-36) questionnaires).[35 36]
- ► Foetal outcomes:
  - – Foetal weight.
  - – Gestational age at delivery.

## Measurements

The full schedule of assessments can be found in figure 3.

### Physical measurements

Height, weight and blood pressure will be measured using standardised calibrated equipment in antenatal clinic.

### Blood samples

Fasting venous blood samples will be collected to assess maternal HbA1c, fasting glucose, insulin, beta-hydroxybutyrate, liver function tests, free fatty acids, thyroid function tests and full blood count. At the end of the study, all samples will be disposed of in accordance with the Human Tissue Act (2004).

### Questionnaires

Participants will be asked to complete four questionnaires at four time points throughout the trial (self-reported). Quality of life and health status will be assessed using the WHO Quality of Life Questionnaire (brief version) and the 36-Item Short Form Survey, respectively.[35 36] Physical activity will be measured using the IPAQ—Short Form, and diet quality will be assessed using the UK Diabetes and Diet Questionnaire.[37 38] These questionnaires are self-reported by participants and have been chosen as they are widely used and validated tools.

### Food diaries

4-day dietary records will be completed using Libro (Nutritics Mobile Application) or paper food diaries, which will be entered into Nutritics software (Nutritics, Dublin, Ireland).[39] Participants who wish to use Libro will receive one-to-one training to use this by the study dietitian. Diaries will provide the research team with information about the intake of energy, carbohydrate, fat, protein, fibre, GI and the timing of meals for participants in both groups. Participants will be asked what other dietary modifications, if any, they have made at their fortnightly dietitian reviews to establish the adoption of any alternative dietary practices in the cohort.

### Adverse events

Participants in both groups will be asked about any adverse effects that they have experienced at each visit. These will include, but are not limited to, the potential effects of a low-energy diet, for example, headache, lethargy, dizziness, constipation, indigestion, poor concentration and hunger. Adverse events will be graded as per CTCAE V.5.[34] Participants will be issued with a participation/emergency card with emergency contact details for the research team to be carried at all times and to be shown to the attending physician in case of emergency admission to hospital. All participants will be issued with clear instructions as to how to manage a hypoglycaemic and/or ketonaemic event (online supplemental file 5).

### Data management

Participant data will be anonymised and will be stored in line with the Medicines for Human Use (Clinical Trials) Amended Regulations 2006 and the Data Protection Act (2018) and archived in line with the Medicines for Human Use (Clinical Trials) Amended Regulations (2006) as defined in the MFT Clinical Trials Office Archiving SOP (11; Retention of Data, Off-Site Archiving, and Destroying Documents). Deidentified data will be stored in a study-specific Research Electronic Data Capture database. The sponsor will periodically audit the site study file, a sample of the case report form, consent forms and source data, and check accuracy of the study database to ensure satisfactory completion.

### Statistical methods

A statistical analysis plan specifying the full details of the primary and secondary outcomes, other variables and methods will be produced prior to trial analysis. The main analysis will be conducted via intention-to-treat population and will not undertake any significance tests. Descriptive, graphical (summary) and basic statistics (eg, (1) number, frequencies and percentages, (2) mean and SD, or (3) median and quartiles as appropriate) will be

| | Study Visit | | | | | | | |
|---|---|---|---|---|---|---|---|---|
| | 1 | 2 | 3 | 4 | 5 | 6 | 7 | 8 |
| Gestation (weeks) | ~24-30 | ~24-30 | ~30-34 | ~32-36 | ~34-38 | ~36-40 | delivery | 11-13 post-partum |
| Eligibility confirmed | X | | | | | | | |
| Informed consent | X | | | | | | | |
| Randomisation | X | | | | | | | |
| Tailored dietitian review (face to face or remote) | X | X | X | X | X | X | | X |
| Height | X | | | | | | | |
| Weight ^ | X | X | X | X | X | X | X | X |
| Blood Pressure ^ | X | X | X | X | X | X | X | X |
| Fasting blood sample * | X | | | | X | | | X |
| Questionnaires # | X | | X | | X | | | X |
| 4-day food diary | | X | | | X | | | X |
| Foetal growth scan | X | | X | | X | | | |
| Review of glucose and ketone measurements | | X | X | X | X | X | | |
| Neonatal measurements § | | | | | | | X | |
| Oral glucose tolerance test | | | | | | | | X |
| Exit interview / end of study questionnaires | | | | | | | | X |
| Invitation to optional qualitative sub-study $ | | | | | | | | X |

^ Frequency of assessment will be 2-4 weekly depending on whether appointment is face-to-face or virtual due to COVID-19 restrictions.

\* Fasting bloods: urea and electrolytes, liver function tests, bone profile, lipids, thyroid function tests, HbA1c, beta-hydroxybutyrate, free fatty acids, full blood count, fasting glucose, insulin

# Questionnaires: World Health Organisation Quality of Life (brief version), 36-Item Short Form Survey, International Physical Activity Questionnaire (short form), UK Diabetes and Diet Questionnaire

§ Neonatal measurements include gestational age at delivery, mode of delivery, and neonatal weight

$ Sub-study involves semi-structured interviews exploring thoughts and experiences of the trial

**Figure 3**   Schedule of assessments.

**Table 1** Trial progression criterion

| | Feasible (green) | Feasible with modification of the protocol (amber) | Not feasible (red) |
|---|---|---|---|
| Recruitment | ≥4 patients/month | >2 patients/month | ≤2 patients/month |
| Uptake to the feasibility study | ≥15% | 10%–15% | <10% |
| Retention to the feasibility study | >70% | 50%–70% | <50% |
| Adherence to the ILED intervention | >50% of the low-energy days completed (2/week between weeks 24-30 and delivery) | 30%–50% of the low-energy days completed (2/week between weeks 24-30 and delivery) | <30% of the low-energy days completed (2/week between weeks 24-30 and delivery) |

ILED, intermittent low-energy diets.

presented as appropriate for each group, respectively, for group difference jointly, and for each stratum. Per-protocol analysis will be considered as a secondary analysis. Levels of missing data will be investigated and used to inform future studies. No imputation will be used. The end of study questionnaire will be analysed using appropriate descriptive statistics for closed questions and key themes will be extracted without formal analysis from open questions to inform future research.

### Progression criterion

The success of the feasibility trial will be defined by the progression criteria as outlined in table 1. Any concerns regarding a low retention rate will be discussed with the PPIE group. Interviews will include those who withdraw from the study to address potential reasons for withdrawal with the aim to improve retention in future.

### Qualitative substudy

Participants will be invited to take part in an optional qualitative substudy at 11–13 weeks post partum. HCPs delivering the interventions will also be invited to take part in this study.

We will undertake semistructured interviews with a subset of women from each group (ILED n=10 and best NHS Care n=10) at around 12 weeks post delivery. The final sample size will be contingent on obtaining data saturation. We will also interview a sample of HCPs involved in the delivery of care to study participants, including dieticians, obstetricians and midwives, including those with leadership and clinical managerial roles. Sampling will be purposive, aiming to obtain women from a range of ethnic groups, ages, socioeconomic backgrounds and self-reported engagement with the intervention. Participants and HCPs will be asked about their experiences and thoughts regarding the intervention, including motivating factors, and facilitators/barriers to engagement. Interviews will be conducted by a researcher from the University of Manchester/MFT who is independent of the research staff involved in the delivery and assessment of the programmes. Analysis will be conducted by two independent researchers at the University of Manchester/

MFT using Braun and Clarke's thematic analysis approach to identify key issues around the acceptability, usefulness of the programmes, and feasibility of a subsequent trial.[40] Analysis will be inductive: open-ended, exploratory and driven by the data.

All participants will also be asked to complete an optional and anonymous end of study questionnaire developed by the study team at their postpartum visit (online supplemental file 6). This will give participants the opportunity to feedback on their experience and will enable the study team to identify improvements to the design of a possible follow-up study.

### Trial steering committee (TSC)

The TSC will include an independent consultant endocrinologist, obstetrician, dietitian and the patient representative. The committee will oversee the trial to ensure that it is carried out to the expected standards. The TSC will liaise with the CI to develop a schedule of meetings, proposed to occur every 4 months, with meetings to occur no less than annually. Minutes will be taken at TSC meetings and copies of the minutes will be filed in the Trial Master File; they will be shared with relevant stakeholders as appropriate.

### Patient and public involvement

Patient and public involvement was actively sought throughout the planning and design of this trial and continues to form a key part of the trial as it progresses. The PPIE group assisted in the development of all participant materials and provided valuable insight into the wording of participant information and acceptability of the proposed intervention. The PPIE group will be updated as the trial progresses and a further focus group will be held to advise on the interview schedule and wording for the qualitative substudy. The group will also be invited to aid in the development of summarising key findings for dissemination to relevant patient groups.

## ETHICS AND DISSEMINATION

This study has been approved by the Cambridge East Research Ethics Committee and is sponsored by MFT. Findings will be disseminated via publication in peer-reviewed journals, conference presentations and shared with diabetes charitable bodies and organisations in the UK, such as Diabetes UK and the Association of British Clinical Diabetologists. Anonymised data will be available on formal request once the principal results of the study have been published. Planned modifications to the protocol will be approved by the research ethics committee before they are adopted into the study. An audit trail of ethical amendments and documentation will allow monitoring by the research team and external regulatory bodies.

This is the first study to assess the feasibility and safety of an ILED in GDM as compared with best NHS care. Given the increasing incidence of GDM and associated health risks, this research is both pertinent and important. The study is not powered to show differences between ILED and best NHS care; however, the planned quantitative and qualitative assessments will inform the feasibility of the programme and a future definitive trial.

### Author affiliations
[1]The University of Manchester, Manchester, UK
[2]Manchester University NHS Foundation Trust, Manchester, UK
[3]Diabetes, Endocrinology and Metabolic Services, Manchester University NHS Foundation Trust, Manchester, UK
[4]Division of Cancer Sciences, The University of Manchester, Manchester, UK
[5]Department of Nutrition and Dietetics, Manchester Foundation Trust, Manchester, UK
[6]Division of Dentistry, University of Manchester, Manchester, UK
[7]Centre for Primary Care and Health Services Research, The University of Manchester, Manchester, UK
[8]University Hospitals of North Midlands NHS Trust, Stoke-on-Trent, UK
[9]Department of Health Professions, Faculty of Health and Education, Manchester Metropolitan University, Manchester, UK

**Acknowledgements** With thanks to Rebecca Lumsden, our patient expert, whose insight has been invaluable in the design of the study. We are most grateful to Jodie Aspinall, and Lisa Brew-Butler, specialist midwives who continue to help identify suitable candidates for the study, and Jessy Athirampuzha, Amanda Hulme, and Jane Davies, who played a critical role in the recruitment and follow-up of participants throughout the trial.

**Contributors** MH, BGI and ED designed the study, wrote the protocol and secured funding. BM designed and worded the qualitative substudy. T-LS developed the statistical analysis plan. FH reviewed and advised on overall study design. AP provided expert obstetric guidance for the protocol design. AV, CL and MH conducted dietetic reviews. WM helped with recruitment and review of participants. JY and BE were responsible for project management and data reporting. SM coordinated the clinical trial. ED drafted the manuscript for publication, with input from MH, BGI, BM and T-LS. All authors have proofed and checked the manuscript.

**Funding** This trial is funded by the National Institute for Health Research (NIHR201944) and sponsored by Manchester University NHS Foundation Trust (MFT). The funders of the study had no role in the study design or writing of the report. Dr Dapre is an NIHR sponsored GP academic clinical fellow.

**Competing interests** Michelle Harvie has coauthored three self-help books for the public to follow intermittent diets. All author proceeds are paid directly to the charity Prevent Breast Cancer (registered charity number 1109839) to fund breast cancer research.

**Patient and public involvement** Patients and/or the public were involved in the design, or conduct, or reporting, or dissemination plans of this research. Refer to the Methods section for further details.

**Patient consent for publication** Not applicable.

**Provenance and peer review** Not commissioned; externally peer reviewed.

**ORCID iDs**
Elizabeth Dapre http://orcid.org/0000-0001-8220-4514
Michelle Harvie http://orcid.org/0000-0001-9761-3089
Brian McMillan http://orcid.org/0000-0002-0683-3877

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
