## [Reviewer comments · BMJ Open]

ARTICLE DETAILS

TITLE (PROVISIONAL)	Manchester Intermittent Diet in Gestational Diabetes Acceptability Study (MIDDAS-GDM): A Two-Arm Randomised Feasibility Protocol Trial of an Intermittent Low-Energy Diet (ILED) in women with Gestational Diabetes and Obesity in Greater Manchester
AUTHORS	Dapre, Elizabeth; Issa, Basil; Harvie, Michelle; Su, Ting-Li; McMillan, Brian; Pilkington, A.; Hanna, F.; Vyas, Avni; Mackie, S.; Yates, James; Evans, Benjamin; Mubita, Womba; Lombardelli, Cheryl

VERSION 1 – REVIEW

REVIEWER	Greaves, Colin University of Birmingham School of Sport Exercise and Rehabilitation Sciences, School of Sport, Exercise and Rehabilitation Science
REVIEW RETURNED	29-Aug-2023

GENERAL COMMENTS	General Comment: This is a very clear and well-written paper and so I have only a few comments. The topic is an important one as gestational diabetes remains a pressing problem for the NHS, both due to its impact on gestational outcomes and future risk of type 2 diabetes. Hence, a pragmatic approach that is effective and deliverable /easy to implement would be highly desirable. One important comment would be to consider planning a implementation-focused process evaluation for the main trial that is to follow (i.e. in parallel to the trial, conduct mixed methods research to find ways to optimise future delivery across a range of NHS settings - interviewing health professionals and service managers involved in delivering the intervention as well as patients). Abstract: Nice and clear - is the date Nov 22 correct? In which case, use of the future tense seems questionable? Background: Very clear and concise rationale. no comments Methods: Clear enough to replicate. Consider reducing the amount of acronyms (MFT etc) which make it harder to read (esp. for non-specialists) The control arm intervention seems over-specified - e.g. "Participants will receive ongoing dietetic education and support every 2 weeks until delivery". This seems much more intensive than maternity services would usually deliver (in most places) and so risks undermining the aim of the trial by giving an active (beyond usual care) intervention to the control arm. I would advise
--

	to use "usual care" as the control treatment (and document what that consists of). That will give you a much better chance of showing that the proposed intervention is better than usual care (which is what matters in terms of informing practice), as opposed to "is the proposed treatment better than some idealised, never-achieved standard of care". This change will also make your intervention more deliverable in practice (just add the 5:2 advice element (and some realistically achievable level of support) to usual care). Sorry, this is a comment on the protocol rather than the paper, but may be worth considering? One important methodological issue - a feasibility study should not be designed " to assess the impact of the intervention on each of the outcome measures" and can never achieve this aim (certainly not with a sample size of 24 per group) - please remove this from the text on sample size. No comparison between groups should be intended or attempted. In the statistical /analysis section need to remove the text "for group difference jointly" for the same reason. In a similar vein, it makes no sense to talk about primary and secondary outcomes or analyses in the context of a feasibility study. Also it makes little sense to have a list of primary outcomes (by definition there would only be one). Why not call them (all) "Feasibility study outcomes" instead? I am slightly concerned about the intention that "Dietary changes in both groups will be assessed". It is OK to report the dietary changes with means and CIs, just to show feasibility of completion and possible sensitivity to change of the measures used (dietary measures can be very flaky in particular), but the robustness of any estimates of change scores is going to be low, so be careful how this is presented (don't hang your hats on it). The same applies to the "exploratory outcomes" listed. UKDDQ is a good choice for dietary measurement (as well as being a good guide for healthy dietary change - indeed it could be considered to be part of the intervention?) Given the 5:2 dietary protocol, does it make sense to use a 4-day food diary? I would at least include a question on "how many days in the last week did you use a food supplement (e.g. nutrient shake)" to get at fidelity to your intended eating protocol. Qualitative section: Is "110-120 semi-structured interviews" a typo? this is far too many interviews (and indeed would go beyond the proposed sample size. I would suggest (feel free to ignore) interviewing the women pre-partum as the mothers' priorities post-partum are likely to be very far removed from taking part in a research study (this seems to be the experience from a number of prior GD studies). Maybe ask your PPI group for advice about this? Again, this is a comment on the protocol rather than the paper, but may be worth considering as a minor ethics amendment? Maybe ask the women when they would prefer to be interviewed /give them the choice to maximise uptake of the interviews?
--	---

REVIEWER	Al Hashmi, Iman Sultan Qaboos University
REVIEW RETURNED	08-Oct-2023

GENERAL COMMENTS

Dear Authors,

We appreciate your submission of your manuscript to the BMJ. The topic of your study is crucial for ensuring quality antenatal care and positive pregnancy outcomes, and I have reviewed your protocol with great interest. Below, I have provided feedback for each section of your manuscript.

Title:

Please ensure consistency between the study title, study abstract, and the main aim in the text. This should include clarifying that the study aims to test feasibility, acceptability and safety.

We recommend adding "two-arm RCT" to the title for clarity.

Introduction:

In reference to NICE, please provide an expanded explanation the first time it is mentioned.

On page 5, line #15, define your concept of "normal eating."

On page 5, line #60, please include a reference to support the statement.

Aim:

We suggest formulating a concise main aim that aligns with the title, abstract, and the rest of the document. Follow this with specific objectives listed in bullet points.

Remove descriptions of study outcomes from this section, as they will be detailed in the "Outcome Measurements" section of the Methods.

Methods:

On page 6, line #20, provide the full name of NHS the first time it appears.

On page 6, line #37, expand "MFT" for clarity.

Include a separate section that describes the standard care provided for women with GDM in the study setting.

In the "Trial Setting and Recruitment" section on page 6, include the expected start date for recruitment.

Consider using random selection from a larger population to enhance the validity of your study results and align with the description of an RCT. Otherwise, clarify that the study is quasi-experimental and comparative.

In the "Trial Setting and Recruitment" section on page 6, provide a detailed description of the study settings, including urban vs. rural, the number of GDM women attending antenatal clinics, types of healthcare providers involved in GDM care, ethics oversight, research and healthcare infrastructure, etc.

Create a subsection under the Methods that describes the study population visiting the study settings.

Provide information for sample size calculation, including power percentage, effect size, alpha level, and specify whether it's one-sided or two-sided.

Justify the selection of different BMI criteria for the general sample and high-risk sample, supported by evidence.

Provide a rationale for choosing age and BMI as stratification factors, supported by evidence.

	On page 8, line #18, provide a clear description of the clinicians who will be involved in data collection to ensure intervention delivery consistency. Simplify the patient information sheet using bullet points and straightforward language to improve readability. On page 10, for Safety Outcomes, add "as measured by participants" to the end of the first two sentences. Consider including potential side effects like the incidence of shoulder dystocia in Fetal Outcomes on pages 10 and 11. On page 10, line #15, describe any planned measures to improve the retention rate, if applicable. Consider using a valid and reliable tool to assess adherence to the recommended diet plan on page 12, line #31, instead of relying solely on self-reporting due to potential bias. Provide a detailed description of the study instruments, including their validity and reliability, and clarify whether questionnaires will be self-reported or completed with assistance from data collectors. On page 27, line #38, explain whether there are plans to train participants on recording their data in the Libro App. At the end of the protocol, please consider including the following: Authors' contribution statement Sponsor name, role, and contact details Thank you for considering my feedback, and I look forward to reviewing the revised manuscript. Sincerely,
--	--

REVIEWER	Feghali, MN University of Pittsburgh
REVIEW RETURNED	11-Oct-2023

GENERAL COMMENTS	Clear description of ongoing study with appropriate methodology and planned analyses. The manuscript is aligned with the study protocol. The planned study outcomes are appropriate for a feasibility trial.
--

VERSION 1 – AUTHOR RESPONSE

Referee #1 (Comments to the Author):

1. One important comment would be to consider planning an implementation-focused process evaluation for the main trial that is to follow (i.e. in parallel to the trial, conduct mixed methods research to find ways to optimise future delivery across a range of NHS settings - interviewing health professionals and service managers involved in delivering the intervention as well as patients). Thank you for raising this. We plan to interview healthcare professionals involved in the study and in the delivery of the diabetes antenatal service, some of whom have leadership and clinical managerial roles (for example, diabetes speciality leads / antenatal clinic nurse manager) for the service and who would be able to provide their views about the potential for the implementation of this intervention in the future. We have clarified this in lines 529-530.

2. Abstract: Nice and clear - is the date Nov 22 correct? In which case, use of the future tense seems questionable? Yes, recruitment began in November 2022. We were advised to use the future tense in

a protocol paper.

3. Methods: Clear enough to replicate. Consider reducing the amount of acronyms (MFT etc) which make it harder to read (esp. for non-specialists). Thank you, we have removed MFT.

4. The control arm intervention seems over-specified - e.g. "Participants will receive ongoing dietetic education and support every 2 weeks until delivery". This seems much more intensive than maternity services would usually deliver (in most places) and so risks undermining the aim of the trial by giving an active (beyond usual care) intervention to the control arm. I would advise to use "usual care" as the control treatment (and document what that consists of). That will give you a much better chance of showing that the proposed intervention is better than usual care (which is what matters in terms of informing practice), as opposed to "is the proposed treatment better than some idealised, never-achieved standard of care". This change will also make your intervention more deliverable in practice (just add the 5:2 advice element (and some realistically achievable level of support) to usual care). Sorry, this is a comment on the protocol rather than the paper, but may be worth considering? Thank you for this comment. This is an important point. Since the amount of dietary support is key to influencing dietary adherence and the success of a dietary intervention, the study is designed so that level of support is matched between the 2 groups. This enables us to see whether any differences in dietary adherence and potential benefits of the diet are specifically associated with the intermittent diet and not the levels of dietetic support.

We acknowledge that GDM dietetic management is often less frequent than our control group but considered the 2 weekly remote reviews to be feasible in delivery of these services and a model of best NHS care. We have clarified this in the best NHS care description (lines 308-310).

5. One important methodological issue - a feasibility study should not be designed "to assess the impact of the intervention on each of the outcome measures" and can never achieve this aim (certainly not with a sample size of 24 per group) - please remove this from the text on sample size. No comparison between groups should be intended or attempted. In the statistical /analysis section need to remove the text "for group difference jointly" for the same reason. Thank you for this comment. This has been removed from the Sample Size text. Following review by our statistician 'for group difference jointly' has been left in the Statistical methods section (line 506) because it is being done in a descriptive sense and is deemed important for future definitive trial planning in terms of indicative effect size / variability.

6. In a similar vein, it makes no sense to talk about primary and secondary outcomes or analyses in the context of a feasibility study. Also it makes little sense to have a list of primary outcomes (by definition there would only be one). Why not call them (all) "Feasibility study outcomes" instead? Thank you; the primary and secondary outcomes (and their associated level of detail) were requested by the funding body and the trial registry, thus we are limited in our capacity to change this wording.

7. I am slightly concerned about the intention that "Dietary changes in both groups will be assessed". It is OK to report the dietary changes with means and CIs, just to show feasibility of completion and possible sensitivity to change of the measures used (dietary measures can be very flaky in particular), but the robustness of any estimates of change scores is going to be low, so be careful how this is presented (don't hang your hats on it). The same applies to the "exploratory outcomes" listed. This was included in error and has now been removed.

8. UKDDQ is a good choice for dietary measurement (as well as being a good guide for healthy dietary change - indeed it could be considered to be part of the intervention?) Thank you for this comment.

9. Given the 5:2 dietary protocol, does it make sense to use a 4-day food diary? I would at least include a question on "how many days in the last week did you use a food supplement (e.g. nutrient shake)" to get at fidelity to your intended eating protocol. Thank you; the length of the food diary was considered. We decided upon a 4 day diary to include 1 low calorie day and 3 'normal' eating days. This is to ensure sufficient dietary information whilst not overburdening participants. The food diary includes a question about food supplements.

10. Qualitative section: Is "110-120 semi-structured interviews" a typo? this is far too many interviews (and indeed would go beyond the proposed sample size. Thanks for pointing this out, this is a typo and has been amended.

11. I would suggest (feel free to ignore) interviewing the women pre-partum as the mothers' priorities post-partum are likely to be very far removed from taking part in a research study (this seems to be the experience from a number of prior GD studies). Maybe ask your PPI group for advice about this? Again, this is a comment on the protocol rather than the paper, but may be worth considering as a minor ethics amendment? Maybe ask the women when they would prefer to be interviewed /give them the choice to maximise uptake of the interviews? Thank you for this query; this is certainly something we will take back to our PPIE group. The feedback so far has been that women are very overburdened with appointments during their pregnancy and as such additional appointments can be off-putting. Additionally, the purpose of the interviews is to determine how women found taking part in the trial and we may miss information if they are held too early. Having discussed your comments we plan to conduct interviews at the same time as Visit 8 (their final oral glucose tolerance test); this appointment is normally around 2-3 hours and therefore presents an ideal opportunity to carry out the interviews between tests.

Referee #2 (Comments to the Author):

Title:

1. Please ensure consistency between the study title, study abstract, and the main aim in the text. This should include clarifying that the study aims to test feasibility, acceptability and safety. Thank you, this has now been updated.
2. We recommend adding "two-arm RCT" to the title for clarity. This has now been updated.

Introduction:

1. In reference to NICE, please provide an expanded explanation the first time it is mentioned. This has now been updated.
2. On page 5, line #15, define your concept of "normal eating." This has now been updated (line 143).
3. On page 5, line #60, please include a reference to support the statement. This wording has been updated; the statement was based on our own PPIE work (line 169-170).

Aim:

1. We suggest formulating a concise main aim that aligns with the title, abstract, and the rest of the document. Follow this with specific objectives listed in bullet points. Thank you for raising this. This has now been reworded (line 174).

2. Remove descriptions of study outcomes from this section, as they will be detailed in the "Outcome Measurements" section of the Methods. These have been removed.

Methods:

1. On page 6, line #20, provide the full name of NHS the first time it appears. Thank you, we have instead provided the full name of the NHS under the Aims section above as this is the first time it appears. We hope this will be acceptable.
2. On page 6, line #37, expand "MFT" for clarity. This has now been updated (line 191).

3. Include a separate section that describes the standard care provided for women with GDM in the study setting. Thank you; we have updated the Study Arm 1 description to make it clear that this is best NHS care (line 308). This level of support is higher than typically provided in NHS GDM antenatal clinics due to limited resources but has been utilised to reflect best NHS care.
4. In the "Trial Setting and Recruitment" section on page 6, include the expected start date for recruitment. This has now been updated (line 191).
5. Consider using random selection from a larger population to enhance the validity of your study results and align with the description of an RCT. Otherwise, clarify that the study is quasi-experimental and comparative. Thank you for this comment. As a feasibility study this trial is not powered to formally evaluate the effects of the intervention. The 'Sample Size' has been reworded to provide detail of the power calculation for the feasibility outcomes (line 242).
6. In the "Trial Setting and Recruitment" section on page 6, provide a detailed description of the study settings, including urban vs. rural, the number of GDM women attending antenatal clinics, types of healthcare providers involved in GDM care, ethics oversight, research and healthcare infrastructure, etc. Thank you, we have updated this section to include more information (line 192). We have not included information on ethics and research infrastructure here as this is covered elsewhere.
7. Create a subsection under the Methods that describes the study population visiting the study settings. We have now included this within Trial Setting and Recruitment (line 192).
8. Provide information for sample size calculation, including power percentage, effect size, alpha level, and specify whether it's one-sided or two-sided. Thank you; this information can be found under Sample Size (line 235) and has been reworded to try to make it clearer. As a feasibility study the measurements cannot be powered. Further detail on statistical methods can be found from line 496.
9. Justify the selection of different BMI criteria for the general sample and high-risk sample, supported by evidence. This is standard NICE guidance; an appropriate reference has now been added (Ref 30: Teare et al).
10. Provide a rationale for choosing age and BMI as stratification factors, supported by evidence. We have now included an appropriate reference (Ref 31: Makgoba et al).
11. On page 8, line #18, provide a clear description of the clinicians who will be involved in data collection to ensure intervention delivery consistency. We have now included a description for clarity (line 285).
12. Simplify the patient information sheet using bullet points and straightforward language to improve readability. Thank you for this comment. We are unable to change the patient information sheet as this has been approved by ethics and is currently in use. We hope that you will be reassured that the design of all patient materials was done in collaboration with, and approved by, our PPIE advisory group.
13. On page 10, for Safety Outcomes, add "as measured by participants" to the end of the first two sentences. This has now been updated.
14. Consider including potential side effects like the incidence of shoulder dystocia in Fetal Outcomes on pages 10 and 11. Thank you for this suggestion. Following discussion we decided against the inclusion of listing side effects because the list of potential labour complications in GDM is extensive. We have instead elaborated on the final bullet point regarding 'the incidence and rate of other adverse effects' under 'safety outcomes' (line 386).
15. On page 10, line #15, describe any planned measures to improve the retention rate, if applicable. Thank you. As a feasibility trial we will be looking at this in our analysis. We are working alongside our PPIE advisory group to address any concerns around retention, we have introduced a congratulatory card for participants following birth, and a continuity call between delivery and final visit. Our qualitative substudy interviews aim to include those who have withdrawn from the study in order to address potential reasons why participants withdraw, and to consider ways to improve retention in a future definitive RCT. We have included further information from line 516.
16. Consider using a valid and reliable tool to assess adherence to the recommended diet plan on page 12, line #31, instead of relying solely on self-reporting due to potential bias. Thank you for your suggestion; we acknowledge that assessing diet adherence is challenging however self-report is the

accepted method and widely used in dietary intervention research. We are trying to assess adherence to dietary guidelines and as such self-reporting is the most appropriate method.

17. Provide a detailed description of the study instruments, including their validity and reliability, and clarify whether questionnaires will be self-reported or completed with assistance from data collectors. These questionnaires were chosen as they are widely used and accepted in diabetes care and research in general. We have now clarified they are self-reported throughout the report.

18. On page 27, line #38, explain whether there are plans to train participants on recording their data in the Libro App. Participants who wish to use Libro will receive one to one training to use this by the study dietitian. We have now clarified this on line 468.

At the end of the protocol, please consider including the following:

1. Authors' contribution statement. This is available from line 740.
2. Sponsor name, role, and contact details. This is available from line 753.

VERSION 2 – REVIEW

REVIEWER	Greaves, Colin University of Birmingham School of Sport Exercise and Rehabilitation Sciences, School of Sport, Exercise and Rehabilitation Science
REVIEW RETURNED	20-Nov-2023

GENERAL COMMENTS	I don't have any problems with the revised text in terms of describing what you intend to do. However, I think the response to my review of the intended research methods is overly dismissive /defensive, rather than taking on board comments that could improve the robustness and value of the study. In practice that is your choice to make as these comments refer to the actual study protocol, rather than the text of the paper. and the protocol paper must report what you intend to do. In reality, I don't think you are as constrained by the funder (NIHR?) as you seem to think and you actually have the freedom to adapt your protocol in consultation with the study steering group if, on reflection, a more robust approach is available. I wish you the best of luck with this important study, although I fear that the future trial might be undermined by the use of an active control group in particular (see the LOOK-AHEAD study for an example of how over-specifying the control treatment can undermine the potential of trials to deliver practically meaningful results - also see Roland et al (British Medical Journal 1998 Vol. 316 Pages 285-285) for an explanation of pragmatic trials and the value of this approach).
--

REVIEWER	Feghali, MN University of Pittsburgh
REVIEW RETURNED	10-Nov-2023

GENERAL COMMENTS	The authors provide a clear description of their ongoing study with appropriate methodology and planned analyses. The updated manuscript addresses the recommended revisions by the reviewers and it is aligned with the study protocol. The proposed study outcomes are appropriate for a feasibility trial and the supplementary material provided align with the expectations of a protocol manuscript.
--

VERSION 2 – AUTHOR RESPONSE

Reviewer: 3

Dr. MN Feghali, University of Pittsburgh

Comments to the Author:

The authors provide a clear description of their ongoing study with appropriate methodology and planned analyses. The updated manuscript addresses the recommended revisions by the reviewers and it is aligned with the study protocol. The proposed study outcomes are appropriate for a feasibility trial and the supplementary material provided align with the expectations of a protocol manuscript. We thank you for your comments.

Reviewer: 1

Prof. Colin Greaves, University of Birmingham School of Sport Exercise and Rehabilitation Sciences

Comments to the Author:

I don't have any problems with the revised text in terms of describing what you intend to do. However, I think the response to my review of the intended research methods is overly dismissive /defensive, rather than taking on board comments that could improve the robustness and value of the study. In practice that is your choice to make as these comments refer to the actual study protocol, rather than the text of the paper. and the protocol paper must report what you intend to do. In reality, I don't think you are as constrained by the funder (NIHR?) as you seem to think and you actually have the freedom to adapt your protocol in consultation with the study steering group if, on reflection, a more robust approach is available. I wish you the best of luck with this important study, although I fear that the future trial might be undermined by the use of an active control group in particular (see the LOOK-AHEAD study for an example of how over-specifying the control treatment can undermine the potential of trials to deliver practically meaningful results - also see Roland et al (British Medical Journal 1998 Vol. 316 Pages 285-285) for an explanation of pragmatic trials and the value of this approach). Thank you for your comments. We apologise if our response came across as dismissive or defensive, that was not our intention. We agree that this methodology does run the risk of undermining the value of our intervention and your suggestions are most certainly something that we will consider in greater detail for a future trial. At present the study is already underway and changes to the methodology at this stage run the risk of affecting the integrity of our results. We have also considered some practical concerns; for example, there is much literature published already which demonstrates how positively an increased level of dietetic support can influence patient dietary choices and feeding behaviour. When considering how the intermittent low-energy diet (ILED) is likely to be implemented within an NHS setting we recognise the limitations of being able to provide the level of dietetic intervention seen within our study. We therefore felt it important to determine whether the ILED itself could have a beneficial effect on blood glucose levels, thus we needed to reduce the potential for dietetic support as a confounding factor by ensuring equal support across the two groups. We will certainly consider alternative approaches in future based on your suggestions.